# Structured Sparse Regression via Greedy Hard-thresholding

**Prateek Jain**
Microsoft Research India

**Nikhil Rao**
Technicolor

**Inderjit Dhillon**
UT Austin

## Abstract

Several learning applications require solving high-dimensional regression problems where the relevant features belong to a small number of (overlapping) groups. For very large datasets and under standard sparsity constraints, hard thresholding methods have proven to be extremely efficient, but such methods require NP hard projections when dealing with overlapping groups. In this paper, we show that such NP-hard projections can not only be avoided by appealing to submodular optimization, but such methods come with strong theoretical guarantees even in the presence of poorly conditioned data (i.e. say when two features have correlation $\geq 0.99$), which existing analyses cannot handle. These methods exhibit an interesting computation-accuracy trade-off and can be extended to significantly harder problems such as sparse overlapping groups. Experiments on both real and synthetic data validate our claims and demonstrate that the proposed methods are orders of magnitude faster than other greedy and convex relaxation techniques for learning with group-structured sparsity.

## 1 Introduction

High dimensional problems where the regressor belongs to a small number of groups play a critical role in many machine learning and signal processing applications, such as computational biology and multitask learning. In most of these cases, the groups overlap, i.e., the same feature can belong to multiple groups. For example, gene pathways overlap in computational biology applications, and parent-child pairs of wavelet transform coefficients overlap in signal processing applications.

The existing state-of-the-art methods for solving such group sparsity structured regression problems can be categorized into two broad classes: a) convex relaxation based methods , b) iterative hard thresholding (IHT) or greedy methods. In practice, IHT methods tend to be significantly more scalable than the (group-)lasso style methods that solve a convex program. But, these methods require a certain projection operator which in general is NP-hard to compute and often certain simple heuristics are used with relatively weak theoretical guarantees. Moreover, existing guarantees for both classes of methods require relatively restrictive assumptions on the data, like Restricted Isometry Property or variants thereof [2, 7, 16], that are unlikely to hold in most common applications. In fact, even under such settings, the group sparsity based convex programs offer at most polylogarithmic gains over standard sparsity based methods [16].

Concretely, let us consider the following linear model:

$$\boldsymbol{y} = X\boldsymbol{w}^* + \boldsymbol{\beta}, \tag{1}$$

where $\boldsymbol{\beta} \sim N(0, \lambda^2 I)$, $X \in \mathbb{R}^{n \times p}$, each row of $X$ is sampled i.i.d. s.t. $\boldsymbol{x}_i \sim N(0, \Sigma)$, $1 \leq i \leq n$, and $\boldsymbol{w}^*$ is a $k^*$-group sparse vector i.e. $\boldsymbol{w}^*$ can be expressed in terms of only $k^*$ groups, $G_j \subseteq [p]$.

The existing analyses for both convex as well as hard thresholding based methods require $\kappa = \sigma_1/\sigma_p \leq c$, where $c$ is an absolute constant (like say 3) and $\sigma_i$ is the $i$-th largest eigenvalue of $\Sigma$.

This is a significantly restrictive assumption as it requires all the features to be nearly independent of each other. For example, if features 1 and 2 have correlation more than say .99 then the restriction on $\kappa$ required by the existing results do not hold.

Moreover, in this setting (i.e., when $\kappa = O(1)$), the number of samples required to exactly recover $\boldsymbol{w}^*$ (with $\lambda = 0$) is given by: $n = \Omega(s + k^* \log M)$ [16], where $s$ is the maximum support size of a union of $k^*$ groups and $M$ is the number of groups. In contrast, if one were to directly use sparse regression techniques (by ignoring group sparsity altogether) then the number of samples is given by $n = \Omega(s \log p)$. Hence, even in the restricted setting of $\kappa = O(1)$, group-sparse regression improves upon the standard sparse regression only by logarithmic factors.

Greedy, Iterative Hard Thresholding (IHT) methods have been considered for group sparse regression problems, but they involve NP-hard projections onto the constraint set [3]. While this can be circumvented using approximate operations, the guarantees they provide are along the same lines as the ones that exist for convex methods.

In this paper, we show that IHT schemes with approximate projections for the group sparsity problem yield much stronger guarantees. Specifically, our result holds for arbitrarily large $\kappa$, and arbitrary group structures. In particular, using IHT with greedy projections, we show that $n = \Omega\left((s \log \frac{1}{\epsilon} + \kappa^2 k^* \log M) \log \frac{1}{\epsilon}\right)$ samples suffice to recover $\epsilon$-approximatation to $\boldsymbol{w}^*$ when $\lambda = 0$. On the other hand, IHT for standard sparse regression [10] requires $n = \Omega(\kappa^2 s \log p)$. Moreover, for general noise variance $\lambda^2$, our method recovers $\hat{\boldsymbol{w}}$ s.t. $\|\hat{\boldsymbol{w}} - \boldsymbol{w}^*\| \le 2\epsilon + \lambda \cdot \kappa \sqrt{\frac{s + \kappa^2 k^* \log M}{n}}$. On the other hand, the existing state-of-the-art results for IHT for group sparsity [4] guarantees $\|\hat{\boldsymbol{w}} - \boldsymbol{w}^*\| \le \lambda \cdot \sqrt{s + k^* \log M}$ for $\kappa \le 3$, i.e., $\hat{\boldsymbol{w}}$ is not a consistent estimator of $\boldsymbol{w}^*$ even for small condition number $\kappa$.

Our analysis is based on an extension of the sparse regression result by [10] that requires *exact* projections. However, a critical challenge in the case of overlapping groups is the projection onto the set of group-sparse vectors is NP-hard in general. To alleviate this issue, we use the connection between submodularity and overlapping group projections and a greedy selection based projection is at least *good enough*. The main contribution of this work is to carefully use the greedy projection based procedure along with hard thresholding iterates to guarantee the convergence to the global optima as long as enough i.i.d. data points are generated from model (1).

Moreover, the simplicity of our hard thresholding operator allows us to easily extend it to more complicated sparsity structures. In particular, we show that the methods we propose can be generalized to the sparse overlapping group setting, and to hierarchies of (overlapping) groups.

We also provide extensive experiments on both real and synthetic datasets that show that our methods are not only faster than several other approaches, but are also accurate despite performing approximate projections. Indeed, even for poorly-conditioned data, IHT methods are an order of magnitude faster than other greedy and convex methods. We also observe a similar phenomenon when dealing with sparse overlapping groups.

## 1.1   Related Work

Several papers, notably [5] and references therein, have studied convergence properties of IHT methods for sparse signal recovery under standard RIP conditions. [10] generalized the method to settings where RIP does not hold, and also to the low rank matrix recovery setting. [21] used a similar analysis to obtain results for nonlinear models. However, these techniques apply only to cases where exact projections can be performed onto the constraint set. Forward greedy selection schemes for sparse [9] and group sparse [18] constrained programs have been considered previously, where a single group is added at each iteration. The authors in [2] propose a variant of CoSaMP to solve problems that are of interest to us, and again, these methods require exact projections.

Several works have studied approximate projections in the context of IHT [17, 6, 12]. However, these results require that the data satisfies *RIP*-style conditions which typically do not hold in real-world regression problems. Moreover, these analyses do not guarantee a *consistent* estimate of the optimal regressor when the measurements have zero-mean random noise. In contrast, we provide results under a more general RSC/RSS condition, which is weaker [20], and provide crisp rates for the error bounds when the noise in measurements is random.

## 2 Group Iterative Hard Thresholding for Overlapping Groups

In this section, we formally set up the group sparsity constrained optimization problem, and then briefly present the IHT algorithm for the same. Suppose we are given a set of $M$ groups that can arbitrarily overlap $\mathcal{G} = \{G_1, \ldots, G_M\}$, where $G_i \subseteq [p]$. Also, let $\cup_{i=1}^{M} G_i = \{1, 2, \ldots, p\}$. We let $\|\boldsymbol{w}\|$ denote the Euclidean norm of $\boldsymbol{w}$, and supp($\boldsymbol{w}$) denotes the support of $\boldsymbol{w}$. For any vector $\boldsymbol{w} \in \mathbb{R}^p$, [8] defined the overlapping group norm as

$$\|\boldsymbol{w}\|^{\mathcal{G}} := \inf \sum_{i=1}^{M} \|\boldsymbol{a}_{G_i}\| \textbf{ s.t. } \sum_{i=1}^{M} \boldsymbol{a}_{G_i} = \boldsymbol{w}, \text{ supp}(\boldsymbol{a}_{G_i}) \subseteq G_i \tag{2}$$

We also introduce the notion of "group-support" of a vector and its group-$\ell_0$ pseudo-norm:

$$\text{G-supp}(\boldsymbol{w}) := \{i \ s.t. \ \|\boldsymbol{a}_{G_i}\| > 0\}, \qquad \|\boldsymbol{w}\|_0^{\mathcal{G}} := \inf \sum_{i=1}^{M} \mathbb{1}\{\|\boldsymbol{a}_{G_i}\| > 0\}, \tag{3}$$

where $\boldsymbol{a}_{G_i}$ satisfies the constraints of (2). $\mathbb{1}\{\cdot\}$ is the indicator function, taking the value 1 if the condition is satisfied, and 0 otherwise. For a set of groups $G$, supp($G$) = $\{G_i, \ i \in G\}$. Similarly, G-supp($S$) = G-supp($\boldsymbol{w}_S$).

Suppose we are given a function $f : \mathbb{R}^p \to \mathbb{R}$ and $M$ groups $\mathcal{G} = \{G_1, \ldots, G_M\}$. The goal is to solve the following group sparsity structured problem (GS-Opt):

$$\textbf{GS-Opt:} \qquad \min_{\boldsymbol{w}} f(\boldsymbol{w}) \textbf{ s.t. } \|\boldsymbol{w}\|_0^{\mathcal{G}} \leq k \tag{4}$$

$f$ can be thought of as a loss function over the training data, for instance, logistic or least squares loss. In the high dimensional setting, problems of the form (4) are somewhat ill posed and are NP-hard in general. Hence, additional assumptions on the loss function ($f$) are warranted to guarantee a reasonable solution. Here, we focus on problems where $f$ satisfies the restricted strong convexity and smoothness conditions:

**Definition 2.1** (RSC/RSS). *The function $f : \mathbb{R}^p \to \mathbb{R}$ satisfies the restricted strong convexity (RSC) and restricted strong smoothness (RSS) of order $k$, if the following holds:*

$$\alpha_k I \preceq H(\boldsymbol{w}) \preceq L_k I,$$

*where $H(\boldsymbol{w})$ is the Hessian of $f$ at any $\boldsymbol{w} \in \mathbb{R}^p$ s.t. $\|\boldsymbol{w}\|_0^{\mathcal{G}} \leq k$.*

Note that the goal of our algorithms/analysis would be to solve the problem for arbitrary $\alpha_k > 0$ and $L_k < \infty$. In contrast, adapting existing IHT results to this setting lead to results that allow $L_k/\alpha_k$ less than a constant (like say 3).

We are especially interested in the linear model described in (1), and in recovering $\boldsymbol{w}^\star$ consistently (i.e. recover $\boldsymbol{w}^\star$ exactly as $n \to \infty$). To this end, we look to solve the following (non convex) constrained least squares problem

$$\textbf{GS-LS:} \qquad \hat{\boldsymbol{w}} = \arg\min_{\boldsymbol{w}} f(\boldsymbol{w}) := \frac{1}{2n}\|\boldsymbol{y} - \boldsymbol{X}\boldsymbol{w}\|^2 \textbf{ s.t. } \|\boldsymbol{w}\|_0^{\mathcal{G}} \leq k \tag{5}$$

with $k \geq k^*$ being a positive, user defined integer [1]. In this paper, we propose to solve (5) using an Iterative Hard Thresholding (IHT) scheme. IHT methods iteratively take a gradient descent step, and then project the resulting vector ($\boldsymbol{g}$) on to the (non-convex) constraint set of group sparse vectos, i.e.,

$$\boldsymbol{w}_* = P_k^{\mathcal{G}}(\boldsymbol{g}) = \arg\min_{\boldsymbol{w}} \|\boldsymbol{w} - \boldsymbol{g}\|^2 \textbf{ s.t } \|\boldsymbol{w}\|_0^{\mathcal{G}} \leq k \tag{6}$$

Computing the gradient is easy and hence the complexity of the overall algorithm heavily depends on the complexity of performing the aforementioned *projection*. Algorithm 1 details the IHT procedure for the group sparsity problem (4). Throughout the paper we consider the same high-level procedure, but consider different projection operators $\widehat{P}_k^{\mathcal{G}}(\boldsymbol{g})$ for different settings of the problem.

| **Algorithm 1** IHT for Group-sparsity | **Algorithm 2** Greedy Projection |
|---|---|
| 1: **Input :** data $\boldsymbol{y}, \boldsymbol{X}$, parameter $k$, iterations $T$, step size $\eta$ <br> 2: **Initialize:** $t = 0$, $\boldsymbol{w}^0 \in \mathbb{R}^p$ a $k$-group sparse vector <br> 3: **for** t = 1, 2, …, T **do** <br> 4: $\quad \boldsymbol{g}_t = \boldsymbol{w}_t - \eta \nabla f(\boldsymbol{w}_t)$ <br> 5: $\quad \boldsymbol{w}_t = \widehat{P}_k^{\mathcal{G}}(\boldsymbol{g}_t)$ where $\widehat{P}_k^{\mathcal{G}}(\boldsymbol{g}_t)$ performs (approximate) projections <br> 6: **end for** <br> 7: **Output :** $\boldsymbol{w}_T$ | **Require:** $\boldsymbol{g} \in \mathbb{R}^p$, parameter $\tilde{k}$, groups $\mathcal{G}$ <br> 1: $\hat{\boldsymbol{u}} = 0$ , $\boldsymbol{v} = \boldsymbol{g}$, $\widehat{\mathcal{G}} = \{0\}$ <br> 2: **for** $t = 1, 2, \ldots \tilde{k}$ **do** <br> 3: $\quad$ Find $G^\star = \arg\max_{G \in \mathcal{G} \setminus \widehat{\mathcal{G}}} \|\boldsymbol{v}_G\|$ <br> 4: $\quad \widehat{\mathcal{G}} = \widehat{\mathcal{G}} \bigcup G^\star$ <br> 5: $\quad \boldsymbol{v} = \boldsymbol{v} - \boldsymbol{v}_{G^\star}$ <br> 6: $\quad \boldsymbol{u} = \boldsymbol{u} + \boldsymbol{v}_{G^\star}$ <br> 7: **end for** <br> 8: Output $\hat{\boldsymbol{u}} := \widehat{P}_k^{\mathcal{G}}(\boldsymbol{g})$, $\widehat{\mathcal{G}} = \text{supp}(\boldsymbol{u})$ |

## 2.1 Submodular Optimization for General $\mathcal{G}$

Suppose we are given a vector $\boldsymbol{g} \in \mathbb{R}^p$, which needs to be projected onto the constraint set $\|\boldsymbol{u}\|_0^{\mathcal{G}} \leq k$ (see (6)). Solving (6) is NP-hard when $\mathcal{G}$ contains arbitrary overlapping groups. To overcome this, $P_k^{\mathcal{G}}(\cdot)$ can be replaced by an approximate operator $\widehat{P}_k^{\mathcal{G}}(\cdot)$ (step 5 of Algorithm 1). Indeed, the procedure for performing projections reduces to a submodular optimization problem [3], for which the standard greedy procedure can be used (Algorithm 2). For completeness, we detail this in Appendix A, where we also prove the following:

**Lemma 2.2.** *Given an arbitrary vector $\boldsymbol{g} \in \mathbb{R}^p$, suppose we obtain $\hat{\boldsymbol{u}}, \widehat{\mathcal{G}}$ as the output of Algorithm 2 with input $\boldsymbol{g}$ and target group sparsity $\tilde{k}$. Let $\boldsymbol{u}_* = P_k^{\mathcal{G}}(\boldsymbol{g})$ be as defined in (6). Then*

$$\|\hat{\boldsymbol{u}} - \boldsymbol{g}\|^2 \leq e^{-\frac{\tilde{k}}{k}} \|(\boldsymbol{g})_{\text{supp}(\boldsymbol{u}_*)}\|^2 + \|\boldsymbol{u}_* - \boldsymbol{g}\|^2$$

*where $e$ is the base of the natural logarithm.*

Note that the term with the exponent in Lemma 2.2 approaches $0$ as $\tilde{k}$ increases. Increasing $\tilde{k}$ should imply more samples for recovery of $\boldsymbol{w}^*$. Hence, this lemma hints at the possibility of trading off sample complexity for better accuracy, despite the projections being approximate. See Section 3 for more details. Algorithm 2 can be applied to *any* $\mathcal{G}$, and is extremely efficient.

## 2.2 Incorporating Full Corrections

IHT methods can be improved by the incorporation of "corrections" after each projection step. This merely entails adding the following step in Algorithm 1 after step 5:

$$\boldsymbol{w}^t = \arg\min_{\tilde{\boldsymbol{w}}} f(\tilde{\boldsymbol{w}}) \text{ s.t. } \text{supp}(\tilde{\boldsymbol{w}}) = \text{supp}(\widehat{P}_k^{\mathcal{G}}(\boldsymbol{g}_t))$$

When $f(\cdot)$ is the least squares loss as we consider, this step can be solved efficiently using Cholesky decompositions via the backslash operator in MATLAB. We will refer to this procedure as IHT-FC. Fully corrective methods in greedy algorithms typically yield significant improvements, both theoretically and in practice [10].

# 3 Theoretical Performance Bounds

We now provide theoretical guarantees for Algorithm 1 when applied to the overlapping group sparsity problem (4). We then specialize the results for the linear regression model (5).

**Theorem 3.1.** *Let $\boldsymbol{w}^* = \arg\min_{\boldsymbol{w}, \|\boldsymbol{w}^{\mathcal{G}}\|_0 \leq k^*} f(\boldsymbol{w})$ and let $f$ satisfy RSC/RSS with constants $\alpha_{k'}$, $L_{k'}$, respectively (see Definition 2.1). Set $k = 32 \left(\frac{L_{k'}}{\alpha_{k'}}\right)^2 \cdot k^* \log \left(\frac{L_{k'}}{\alpha_{k'}} \cdot \frac{\|\boldsymbol{w}^*\|_2}{\epsilon}\right)$ and let $k' \leq 2k + k^*$. Suppose we run Algorithm 1, with $\eta = 1/L_{k'}$ and projections computed according to Algorithm 2. Then, the following holds after $t + 1$ iterations:*

$$\|\boldsymbol{w}_{t+1} - \boldsymbol{w}^*\|_2 \leq \left(1 - \frac{\alpha_{k'}}{10 \cdot L_{k'}}\right) \cdot \|\boldsymbol{w}_t - \boldsymbol{w}^*\|_2 + \gamma + \frac{\alpha_{k'}}{L_{k'}} \epsilon,$$

where $\gamma = \frac{2}{L_{k'}} \max_{S, \, s.t., \, | \text{G-supp}(S)| \leq k} \|(\nabla f(\boldsymbol{w}^*))_S\|_2$. *Specifically, the output of the* $T = O\left(\frac{L_{k'}}{\alpha_{k'}} \cdot \frac{\|\boldsymbol{w}^*\|_2}{\epsilon}\right)$-*th iteration of Algorithm 1 satisfies:*

$$\|\boldsymbol{w}_T - \boldsymbol{w}^*\|_2 \leq 2\epsilon + \frac{10 \cdot L_{k'}}{\alpha_{k'}} \cdot \gamma.$$

The proof uses the fact that Algorithm 2 performs approximately good projections. The result follows from combining this with results from convex analysis (RSC/RSS) and a careful setting of parameters. We prove this result in Appendix B.

**Remarks**

Theorem 3.1 shows that Algorithm 1 recovers $\boldsymbol{w}^*$ up to $O\left(\frac{L_{k'}}{\alpha_{k'}} \cdot \gamma\right)$ error. If $\|\arg\min_w f(w)\|_0^{\mathcal{G}} \leq k$, then, $\gamma = 0$. In general our result obtains an additive error which is weaker than what one can obtain for a convex optimization problem. However, for typical statistical problems, we show that $\gamma$ is small and gives us nearly optimal statistical generalization error (for example, see Theorem 3.2).

Theorem 3.1 displays an interesting interplay between the desired accuracy $\epsilon$, and the penalty we thus pay as a result of performing approximate projections $\gamma$. Specifically, as $\epsilon$ is made small, $k$ becomes large, and thus so does $\gamma$. Conversely, we can let $\epsilon$ be large so that the projections are coarse, but incur a smaller penalty via the $\gamma$ term. Also, since the projections are not too accurate in this case, we can get away with fewer iterations. Thus, there is a tradeoff between estimation error $\epsilon$ and model selection error $\gamma$. Also, note that the inverse dependence of $k$ on $\epsilon$ is only logarithmic in nature.

We stress that our results *do not hold* for arbitrary approximate projection operators. Our proof critically uses the greedy scheme (Algorithm 2), via Lemma 2.2. Also, as discussed in Section 4, the proof easily extends to other structured sparsity sets that allow such greedy selection steps.

We obtain similar result as [10] for the standard sparsity case, i.e., when the groups are singletons. However, our proof is significantly simpler and allows for a significantly easier setting of $\eta$.

## 3.1 Linear Regression Guarantees

We next proceed to the standard linear regression model considered in (5). To the best of our knowledge, this is the first consistency result for overlapping group sparsity problems, especially when the data can be arbitrarily conditioned. Recall that $\sigma_{\max}$ ($\sigma_{\min}$) are the maximum (minimum) singular value of $\Sigma$, and $\kappa := \sigma_{\max}/\sigma_{min}$ is the condition number of $\Sigma$.

**Theorem 3.2.** *Let the observations* $\boldsymbol{y}$ *follow the model in* (1). *Suppose* $\boldsymbol{w}^*$ *is* $k^*$-*group sparse and let* $f(\boldsymbol{w}) := \frac{1}{2n}\|X\boldsymbol{w} - y\|_2^2$. *Let the number of samples satisfy:*

$$n \geq \Omega\left((s + \kappa^2 \cdot k^* \cdot \log M) \cdot \log\left(\frac{\kappa}{\epsilon}\right)\right),$$

*where* $s = \max_{\boldsymbol{w}, \|\boldsymbol{w}\|_0^{\mathcal{G}} \leq k} |\text{supp}(\boldsymbol{w})|$. *Then, applying Algorithm 1 with* $k = 8\kappa^2 k^* \cdot \log\left(\frac{\kappa}{\epsilon}\right)$, $\eta = 1/(4\sigma_{\max})$, *guarantees the following after* $T = \Omega\left(\kappa \log \frac{\kappa \cdot \|\boldsymbol{w}^*\|_2}{\epsilon}\right)$ *iterations (w.p.* $\geq 1 - 1/n^8$):

$$\|\boldsymbol{w}_T - \boldsymbol{w}^*\| \leq \lambda \cdot \kappa\sqrt{\frac{s + \kappa^2 k^* \log M}{n}} + 2\epsilon$$

**Remarks**

Note that one can ignore the group sparsity constraint, and instead look to recover the (at most) $s$-sparse vector $\boldsymbol{w}^*$ using IHT methods for $\ell_0$ optimization [10]. However, the corresponding sample complexity is $n \geq \kappa^2 s \log(p)$. Hence, for an ill conditioned $\Sigma$, using group sparsity based methods provide a significantly stronger result, especially when the groups overlap significantly.

Note that the number of samples required increases logarithmically with the accuracy $\epsilon$. Theorem 3.2 thus displays an interesting phenomenon: by obtaining more samples, one can provide a smaller recovery error while incurring a larger approximation error (since we choose more groups).

Our proof critically requires that when restricted to group sparse vectors, the least squares objective function $f(\boldsymbol{w}) = \frac{1}{2n}\|\boldsymbol{y} - X\boldsymbol{w}\|_2^2$ is strongly convex as well as strongly smooth:

**Lemma 3.3.** *Let $X \in \mathbb{R}^{n \times p}$ be such that each $x_i \sim \mathcal{N}(0, \Sigma)$. Let $w \in \mathbb{R}^p$ be $k$-group sparse over groups $\mathcal{G} = \{G_1, \ldots G_M\}$, i.e., $\|w\|_0^{\mathcal{G}} \leq k$ and $s = \max_{w, \|w\|_0^{\mathcal{G}} \leq k} |\operatorname{supp}(w)|$. Let the number of samples $n \geq \Omega(C(k \log M + s))$. Then, the following holds with probability $\geq 1 - 1/n^{10}$:*

$$\left(1 - \frac{4}{\sqrt{C}}\right) \sigma_{\min} \|w\|_2^2 \leq \frac{1}{n} \|Xw\|_2^2 \leq \left(1 + \frac{4}{\sqrt{C}}\right) \sigma_{\max} \|w\|_2^2,$$

We prove Lemma 3.3 in Appendix C. Theorem 3.2 then follows by combining Lemma 3.3 with Theorem 3.1. Note that in the least squares case, these are the Restricted Eigenvalue conditions on the matrix $X$, which as explained in [20] are much weaker than traditional RIP assumptions on the data. In particular, RIP requires almost 0 correlation between any two features, while our assumption allows for arbitrary high correlations albeit at the cost of a larger number of samples.

## 3.2 IHT with Exact Projections $P_k^{\mathcal{G}}(\cdot)$

We now consider the setting where $P_k^{\mathcal{G}}(\cdot)$ can be computed exactly and efficiently for any $k$. Examples include the dynamic programming based method by [3] for certain group structures, or Algorithm 2 when the groups do not overlap. Since the exact projection operator can be arbitrary, our proof of Theorem 3.1 does not apply directly in this case. However, we show that by exploiting the structure of hard thresholding, we can still obtain a similar result:

**Theorem 3.4.** *Let $w^* = \arg\min_{w, \|w^{\mathcal{G}}\|_0 \leq k^*} f(w)$. Let $f$ satisfy RSC/RSS with constants $\alpha_{2k+k^*}$, $L_{2k+k^*}$, respectively (see Definition 2.1). Then, the following holds for the $T = O\left(\frac{L_{k'}}{\alpha_{k'}} \cdot \frac{\|w^*\|_2}{\epsilon}\right)$-th iterate of Algorithm 1 (with $\eta = 1/L_{2k+k^*}$) with $\widehat{P}_k^{\mathcal{G}}(\cdot) = P_k^{\mathcal{G}}(\cdot)$ being the exact projection:*

$$\|w_T - w^*\|_2 \leq \epsilon + \frac{10 \cdot L_{k'}}{\alpha_{k'}} \cdot \gamma.$$

*where $k' = 2k + k^*$, $k = O((\frac{L_{k'}}{\alpha_{k'}})^2 \cdot k^*)$, $\gamma = \frac{2}{L_{k'}} \max_{S, s.t., |\text{G-supp}(S)| \leq k} \|(\nabla f(w^*))_S\|_2$.*

See Appendix D for a detailed proof. Note that unlike greedy projection method (see Theorem 3.1), $k$ is independent of $\epsilon$. Also, in the linear model, the above result also leads to consistent estimate of $w^*$.

## 4 Extension to Sparse Overlapping Groups (SoG)

The SoG model generalizes the overlapping group sparse model, allowing the selected groups themselves to be sparse. Given positive integers $k_1, k_2$ and a set of groups $\mathcal{G}$, IHT for SoG would perform projections onto the following set:

$$\mathcal{C}_0^{sog} := \left\{ w = \sum_{i=1}^M a_{G_i} : \|w\|_0^{\mathcal{G}} \leq k_1, \|a_{G_1}\|_0 \leq k_2 \right\} \tag{7}$$

As in the case of overlapping group lasso, projection onto (7) is NP-hard in general. Motivated by our greedy approach in Section 2, we propose a similar method for SoG (see Algorithm 3). The algorithm essentially greedily selects the groups that have large top-$k_2$ elements by magnitude.

Below, we show that the IHT (Algorithm 1) combined with the greedy projection (Algorithm 3) indeed converges to the optimal solution. Moreover, our experiments (Section 5) reveal that this method, when combined with full corrections, yields highly accurate results significantly faster than the state-of-the-art.

We suppose that there exists a set of supports $\mathcal{S}_{k^*}$ such that $\operatorname{supp}(w^*) \in \mathcal{S}_{k^*}$. Then, we obtain the following result, proved in Appendix E:

**Theorem 4.1.** *Let $w^* = \arg\min_{w, \operatorname{supp}(w) \in \mathcal{S}_{k^*}} f(w)$, where $\mathcal{S}_{k^*} \subseteq \mathcal{S}_k \subseteq \{0, 1\}^p$ is a fixed set parameterized by $k^*$. Let $f$ satisfy RSC/RSS with constants $\alpha_k$, $L_k$, respectively. Furthermore, assume that there exists an approximately good projection operator for the set defined in (7) (for example, Algorithm 3). Then, the following holds for the $T = O\left(\frac{L_{k'}}{\alpha_{k'}} \cdot \frac{\|w^*\|_2}{\epsilon}\right)$-th iterate of Algorithm 1 :*

$$\|w_T - w^*\|_2 \leq 2\epsilon + \frac{10 \cdot L_{2k+k^*}}{\alpha_{2k+k^*}} \cdot \gamma,$$

*where $k = O((\frac{L_{2k+k^*}}{\alpha_{2k+k^*}})^2 \cdot k^* \cdot \beta_{\frac{\alpha_{2k+k^*}}{L_{2k+k^*}} \epsilon})$, $\gamma = \frac{2}{L_{2k+k^*}} \max_{S, S \in \mathcal{S}_k} \|(\nabla f(w^*))_S\|_2$.*

---

**Algorithm 3** Greedy Projections for SoG

---

**Require:** $\boldsymbol{g} \in \mathbb{R}^p$, parameters $k_1, k_2$, groups $\mathcal{G}$

  1: $\hat{\boldsymbol{u}} = 0$ , $\boldsymbol{v} = \boldsymbol{g}$, $\widehat{\mathcal{G}} = \{0\}$, $\hat{S} = \{0\}$
  2: **for** t=1,2,…$k_1$ **do**
  3:     Find $G^\star = \arg\max_{G \in \mathcal{G} \backslash \widehat{\mathcal{G}}} \|\boldsymbol{v}_G\|$
  4:     $\widehat{\mathcal{G}} = \widehat{\mathcal{G}} \bigcup G^\star$
  5:     Let $S$ correspond to the indices of the top $k_2$ entries of $\boldsymbol{v}_{G^\star}$ by magnitude
  6:     Define $\bar{\boldsymbol{v}} \in \mathbb{R}^p$, $\bar{\boldsymbol{v}}_S = (\boldsymbol{v}_{G^\star})_S$ $\bar{\boldsymbol{v}}_i = 0$ $i \notin S$
  7:     $\hat{S} = \hat{S} \bigcup S$
  8:     $\boldsymbol{v} = \boldsymbol{v} - \bar{\boldsymbol{v}}$
  9:     $\boldsymbol{u} = \boldsymbol{u} + \bar{\boldsymbol{v}}$
10: **end for**
11: Output $\hat{\boldsymbol{u}}$, $\widehat{\mathcal{G}}$, $\hat{S}$

---

**Remarks**

Similar to Theorem 3.1, we see that there is a tradeoff between obtaining accurate projections $\epsilon$ and model mismatch $\gamma$. Specifically in this case, one can obtain small $\epsilon$ by increasing $k_1, k_2$ in Algorithm 3. However, this will mean we select large number of groups, and subsequently $\gamma$ increases.

A result similar to Theorem 3.2 can be obtained for the case when $f$ is least squares loss function. Specifically, the sample complexity evaluates to $n \geq \kappa^2 \left( k_1^* \log(M) + \kappa^2 k_1^* k_2^* \log(\max_i |G_i|) \right)$. We obtain results for least squares in Appendix F.

An interesting extension to the SoG case is that of a hierarchy of overlapping, sparsely activated groups. When the groups at each level do not overlap, this reduces to the case considered in [11]. However, our theory shows that when a corresponding approximate projection operator is defined for the hierarchical overlapping case (extending Algorithm 3), IHT methods can be used to obtain the solution in an efficient manner.

## 5   Experiments and Results

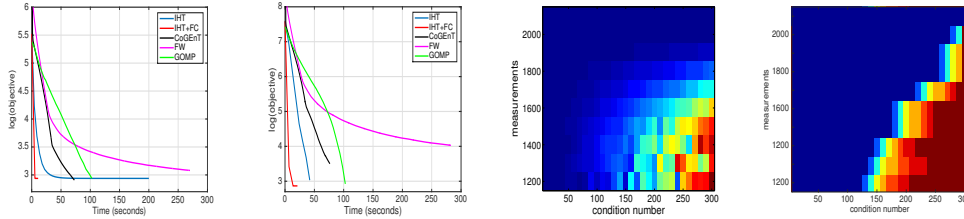

Figure 1: (From left to right) Objective value as a function of time for various methods, when data is well conditioned and poorly conditioned. The latter two figures show the phase transition plots for poorly conditioned data, for IHT and GOMP respectively.

In this section, we empirically compare and contrast our proposed group IHT methods against the existing approaches to solve the overlapping group sparsity problem. At a high level, we observe that our proposed variants of IHT indeed outperforms the existing state-of-the-art methods for group-sparse regression in terms of time complexity. Encouragingly, IHT also performs competitively with the existing methods in terms of accuracy. In fact, our results on the breast cancer dataset shows a 10% relative improvement in accuracy over existing methods.

Greedy methods for group sparsity have been shown to outperform proximal point schemes, and hence we restrict our comparison to greedy procedures. We compared four methods: our algorithm with (**IHT-FC**) and without (**IHT**) the fully corrective step, the Frank Wolfe (**FW**) method [19] , **CoGEnT**, [15] and the Group OMP (**GOMP**) [18]. All relevant hyper-parameters were chosen via a grid search, and experiments were run on a macbook laptop with a 2.5 GHz processor and 16gb memory. Additional experimental results are presented in Appendix G

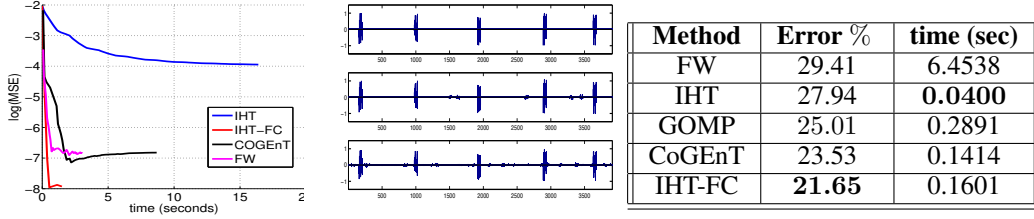

| Method | Error % | time (sec) |
|--------|---------|-----------|
| FW | 29.41 | 6.4538 |
| IHT | 27.94 | **0.0400** |
| GOMP | 25.01 | 0.2891 |
| CoGEnT | 23.53 | 0.1414 |
| IHT-FC | **21.65** | 0.1601 |

Figure 2: (Left) SoG: error vs time comparison for various methods, (Center) SoG: reconstruction of the true signal (top) from IHT-FC (middle) and CoGEnT (bottom). (Right:) Tumor Classification: misclassification rate of various methods.

**Synthetic Data, well conditioned**: We first compared various greedy schemes for solving the overlapping group sparsity problem on synthetic data. We generated $M = 1000$ groups of contiguous indices of size 25; the last 5 entries of one group overlap with the first 5 of the next. We randomly set 50 of these to be active, populated by uniform $[-1, 1]$ entries. This yields $\boldsymbol{w}^\star \in \mathbb{R}^p$, $p \sim 22000$. $\boldsymbol{X} \in \mathbb{R}^{n \times p}$ where $n = 5000$ and $X_{ij} \overset{i.i.d}{\sim} N(0, 1)$. Each measurement is corrupted with Additive White Gaussian Noise (AWGN) with standard deviation $\lambda = 0.1$. IHT mehods achieve orders of magnitude speedup compared to the competing schemes, and achieve almost the same (final) objective function value despite approximate projections (Figure 1 (Left)).

**Synthetic Data, poorly conditioned**: Next, we consider the exact same setup, but with each row of $X$ given by: $\boldsymbol{x}_i \sim N(0, \Sigma)$ where $\kappa = \sigma_{\max}(\Sigma)/\sigma_{\min}(\Sigma) = 10$. Figure 1 (Center-left) shows again the advantages of using IHT methods; IHT-FC is about 10 times faster than the next best CoGEnT.

We next generate phase transition plots for recovery by our method (IHT) as well as the state-of-the-art GOMP method. We generate vectors in the same vein as the above experiment, with $M = 500$, $B = 15$, $k = 25$, $p \sim 5000$. We vary the the condition number of the data covariance ($\Sigma$) as well as the number of measurements ($n$). Figure 1 (Center-right and Right) shows the phase transition plot as the measurements and the condition number are varied for IHT, and GOMP respectively. The results are averaged over 10 independent runs. It can be seen that even for condition numbers as high as 200, $n \sim 1500$ measurements suffices for IHT to exactly recovery $\boldsymbol{w}^*$, whereas GOMP with the same setting is not able to recover $\boldsymbol{w}^*$ even once.

**Tumor Classification, Breast Cancer Dataset**   We next compare the aforementioned methods on a gene selection problem for breast cancer tumor classification. We use the data used in [8] [2]. We ran a 5-fold cross validation scheme to choose parameters, where we varied $\eta \in \{2^{-5}, 2^{-4}, \ldots, 2^3\}$ $k \in \{2, 5, 10, 15, 20, 50, 100\}$ $\tau \in \{2^3, 2^4, \ldots, 2^{13}\}$. Figure 2 (Right) shows that the vanilla hard thresholding method is competitive despite performing approximate projections, and the method with full corrections obtains the best performance among the methods considered. We randomly chose 15% of the data to test on.

**Sparse Overlapping Group Lasso:** Finally, we study the sparse overlapping group (SoG) problem that was introduced and analyzed in [14] (Figure 2). We perform projections as detailed in Algorithm 3. We generated synthetic vectors with 100 groups of size 50 and randomly selected 5 groups to be active, and among the active group only set 30 coefficients to be non zero. The groups themselves were overlapping, with the last 10 entries of one group shared with the first 10 of the next, yielding $p \sim 4000$. We chose the best parameters from a grid, and we set $k = 2k^*$ for the IHT methods.

## 6   Conclusions and Discussion

We proposed a greedy-IHT method that can applied to regression problems over set of group sparse vectors. Our proposed solution is efficient, scalable, and provide fast convergence guarantees under general RSC/RSS style conditions, unlike existing methods. We extended our analysis to handle even more challenging structures like sparse overlapping groups. Our experiments show that IHT methods achieve fast, accurate results even with greedy and approximate projections.

## Footnotes

[1] typically chosen via cross-validation

[2]download at $http://cbio.ensmp.fr/ljacob/$

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
