[Supplementary Material]

# A    Using submodularity to perform projections

While solving (6) is NP-hard in general, the authors in [3] showed that it can be approximately solved using methods from submodular function optimization, which we quickly recap here. First, (6) can be cast in the following equivalent way:

$$\hat{\mathcal{G}} = \arg \max_{|\tilde{\mathcal{G}}| \leq k} \left\{ \sum_{i \in I} \boldsymbol{g}_i^2 : \ I = \cup_{G \in \tilde{\mathcal{G}}} G \right\} \tag{8}$$

Once we have $\hat{\mathcal{G}}$, $\hat{\boldsymbol{u}}$ can be recovered by simply setting $\hat{\boldsymbol{u}}_I = \boldsymbol{g}_I$ and $0$ everywhere else, where $I = \cup_{G \in \hat{\mathcal{G}}} G$. Next, we have the following result

**Lemma A.1.** *Given a set $S \in [p]$, the function $z(S) = \sum_{i \in S} \boldsymbol{x}_i^2$. is submodular.*

*Proof.* First, recall the definition of a submodular function:

**Definition A.2.** *Let $Q$ be a finite set, and let $z(\cdot)$ be a real valued function defined on $\Omega^Q$, the power set of $Q$. The function $z(\cdot)$ is said to be submodular if*

$$z(S) + z(T) \geq z(S \cup T) + z(S \cap T) \ \ \forall S, T \subset \Omega^Q$$

Let $S$ and $T$ be two sets of groups, s.t., $S \subseteq T$. Let, $SS = \text{supp}(\cup_{j \in S} G_j)$ and $TT = \text{supp}(\cup_{j \in T} G_j)$. Then, $SS \subseteq TT$. Hence,

$$z(S \cup i) - z(S) = \sum_{\ell \in SS \cup \text{supp}(G_i)} \boldsymbol{x}_\ell^2 - \sum_{\ell \in SS} \boldsymbol{x}_\ell^2$$

$$= \sum_{\ell \in \text{supp}(G_i) \backslash SS} \boldsymbol{x}_\ell^2 \overset{\zeta_1}{\geq} \sum_{\ell \in \text{supp}(G_i) \backslash TT} \boldsymbol{x}_\ell = z(T \cup i) - z(T),$$

where $\zeta_1$ follows from $SS \subseteq TT$. This completes the proof. $\qquad\square$

This result shows that (8) can be cast as a problem of the form

$$\max_{S \subset Q} z(S), \ \text{s.t.} \ |S| \leq k. \tag{9}$$

Algorithm 2, which details the pseudocode for performing approximate projections, exactly corresponds to the greedy algorithm for submodular optimization [1], and this gives us a means to assess the quality of our projections.

## A.1    Proof of Lemma 2.2

*Proof.* First, from the approximation property of the greedy algorithm [13],

$$\|\hat{\boldsymbol{u}}\|^2 \geq \left(1 - e^{-\frac{k'}{k}}\right) \|\boldsymbol{u}_*\|^2 \tag{10}$$

Also, $\|\boldsymbol{g} - \hat{\boldsymbol{u}}\|^2 = \|\boldsymbol{g}\|^2 - \|\hat{\boldsymbol{u}}\|^2$ because $(\hat{\boldsymbol{u}})_{\text{supp}(\hat{\boldsymbol{u}})} = (\boldsymbol{g})_{\text{supp}(\hat{\boldsymbol{u}})}$ and $0$ otherwise.

Using the above two equations, we have:

$$\begin{aligned}
\|\boldsymbol{g} - \hat{\boldsymbol{u}}\|^2 &\leq \|\boldsymbol{g}\|^2 - \|\boldsymbol{u}_*\|^2 + e^{-\frac{k'}{k}} \|\boldsymbol{u}_*\|^2, \\
&= \|\boldsymbol{g} - \boldsymbol{u}_*\|^2 + e^{-\frac{k'}{k}} \|\boldsymbol{u}_*\|^2, \\
&= \|\boldsymbol{g} - \boldsymbol{u}_*\|^2 + e^{-\frac{k'}{k}} \|(\boldsymbol{g})_{\text{supp}(\boldsymbol{u}_*)}\|^2, \tag{11}
\end{aligned}$$

where both equalities above follow from the fact that due to optimality, $(\boldsymbol{u}_*)_{\text{supp}(\boldsymbol{u}_*)} = (\boldsymbol{g})_{\text{supp}(\boldsymbol{u}_*)}$.

$\qquad\square$

# B Proof of Theorem 3.1

*Proof.* Recall that $\boldsymbol{g}_t = \boldsymbol{w}_t - \eta\nabla f(\boldsymbol{w}_t)$, $\boldsymbol{w}_{t+1} = \widehat{P}_k^{\mathcal{G}}(\boldsymbol{g}_t)$.

Let $\mathrm{supp}(\boldsymbol{w}_{t+1}) = S_{t+1}$, $\mathrm{supp}(\boldsymbol{w}^*) = S_*$, $I = S_{t+1} \cup S_*$, and $M = S_* \backslash S_{t+1}$. Also, note that $|\,\mathrm{G\text{-}supp}(I)| \le k + k^*$.

Moreover, $(\boldsymbol{w}_{t+1})_{S_{t+1}} = (\boldsymbol{g}_t)_{S_{t+1}}$ (See Algorithm 2). Hence, $\|(\boldsymbol{w}_{t+1} - \boldsymbol{g}_t)_{S_{t+1} \cup S_*}\|_2^2 = \|(\boldsymbol{g}_t)_M\|_2^2$.

Now, using Lemma B.2 with $\boldsymbol{z} = (\boldsymbol{g}_t)_I$, we have:

$$\|(\boldsymbol{w}_{t+1} - \boldsymbol{g}_t)_I\|_2^2 = \|(\boldsymbol{g}_t)_M\|_2^2 \overset{\zeta_1}{\le} \frac{k^*}{k - \widetilde{k}} \cdot \|(\boldsymbol{g}_t)_{S_{t+1}\backslash S_*}\|_2^2 + \frac{k^*\epsilon}{k - \widetilde{k}},$$

$$\overset{\zeta_2}{\le} \frac{k^*}{k - \widetilde{k}} \cdot \|(\boldsymbol{w}^* - \boldsymbol{g}_t)_I\|_2^2 + \frac{k^*\epsilon}{k - \widetilde{k}}, \tag{12}$$

where $\zeta_1$ follows from $M \subset S_*$ and hence $|\,\mathrm{G\text{-}supp}(M)| \le |\,\mathrm{G\text{-}supp}(S_*)| = k^*$. $\zeta_2$ follows since $\boldsymbol{w}_{S_{t+1}\backslash S_*}^* = 0$.

Now, using the fact that $\|(\boldsymbol{w}_{t+1} - \boldsymbol{w}^*)_I\|_2 = \|\boldsymbol{w}_{t+1} - \boldsymbol{w}^*\|_2$ along with triangle inequality, we have:

$\|\boldsymbol{w}_{t+1} - \boldsymbol{w}^*\|_2$

$$\le \left(1 + \sqrt{\frac{k^*}{k - \widetilde{k}}}\right) \cdot \|(\boldsymbol{w}^* - \boldsymbol{g}_t)_I\|_2 + \sqrt{\frac{k^*\epsilon}{k - \widetilde{k}}}, \tag{13}$$

$$\overset{\zeta_1}{\le} \left(1 + \sqrt{\frac{k^*}{k - \widetilde{k}}}\right) \cdot \|(\boldsymbol{w}^* - \boldsymbol{w}_t - \eta(\nabla f(\boldsymbol{w}^*) - \nabla f(\boldsymbol{w}_t)))_I\|_2 + 2\eta\|(\nabla f(\boldsymbol{w}^*))_{S_{t+1}}\|_2 + \sqrt{\frac{k^*\epsilon}{k - \widetilde{k}}},$$

$$\overset{\zeta_2}{\le} \left(1 + \sqrt{\frac{k^*}{k - \widetilde{k}}}\right) \cdot \|(I - \eta H_{(I \cup S_t)(I \cup S_t)}(\alpha))(\boldsymbol{w}_t - \boldsymbol{w}^*)_{I \cup S_t}\|_2 + 2\eta\|(\nabla f(\boldsymbol{w}^*))_{S_{t+1}}\|_2 + \sqrt{\frac{k^*\epsilon}{k - \widetilde{k}}},$$

$$\overset{\zeta_3}{\le} \left(1 + \sqrt{\frac{k^*}{k - \widetilde{k}}}\right) \cdot \left(1 - \frac{\alpha_{2k+k^*}}{L_{2k+k^*}}\right)\|\boldsymbol{w}_t - \boldsymbol{w}^*\|_2 + \frac{2}{L_{2k+k^*}}\|(\nabla f(\boldsymbol{w}^*))_{S_{t+1}}\|_2 + \sqrt{\frac{k^*\epsilon}{k - \widetilde{k}}}, \tag{14}$$

where $\alpha = c\boldsymbol{w}_t + (1-c)\boldsymbol{w}^*$ for $c > 0$ and $H(\alpha)$ is the Hessian of $f$ evaluated at $\alpha$. $\zeta_1$ follows from triangle inequality, $\zeta_2$ follows from the Mean-Value theorem and $\zeta_3$ follows from the RSC/RSS condition and by setting $\eta = 1/L_{2k+k^*}$.

The theorem now follows by setting $k = 2\left(\left(\frac{L_{2k+k^*}}{\alpha_{2k+k^*}}\right)^2 + 1\right) \cdot \log(\|\boldsymbol{w}^*\|_2/\epsilon)$ and $\epsilon$ appropriately.

$\square$

**Lemma B.1.** *Let* $\boldsymbol{w} = \widehat{P}_k^{\mathcal{G}}(\boldsymbol{g})$ *and let* $S = \mathrm{supp}(\boldsymbol{w})$. *Then, for every* $I$ *s.t.* $S \subseteq I$, *the following holds:*

$$\boldsymbol{w}_I = \widehat{P}_k^{\mathcal{G}}(\boldsymbol{g}_I).$$

*Proof.* Let $Q = \{i_1, i_2, \ldots, i_k\}$ be the $k$-groups selected when the greedy procedure (Algorithm 2) is applied to $\boldsymbol{g}$. Then,

$$\|\boldsymbol{w}_{G_{i_j} \backslash (\cup_{1 \le \ell \le j-1} G_{i_\ell})}\|_2^2 \ge \|\boldsymbol{w}_{G_i \backslash (\cup_{1 \le \ell \le j-1} G_{i_\ell})}\|_2^2, \ \forall 1 \le j \le k, \ \forall i \notin Q.$$

Moreover, the greedy selection procedure is **deterministic**. Hence, even if groups $G_i$ are restricted to lie in a subset of $\mathcal{G}$, the output of the procedure remains exactly the same. $\square$

**Lemma B.2.** *Let* $\boldsymbol{z} \in \mathbb{R}^p$ *be any vector. Let* $\widehat{\boldsymbol{w}} = \widehat{P}_k^{\mathcal{G}}(\boldsymbol{z})$ *and let* $\boldsymbol{w}^* \in \mathbb{R}^p$ *be s.t.* $|\,\mathrm{G\text{-}supp}(\boldsymbol{w}^*)| \le k^*$. *Let* $S = \mathrm{supp}(\widehat{\boldsymbol{w}})$, $S_* = \mathrm{supp}(\boldsymbol{w}^*)$, $I = S \cup S_*$, *and* $M = S_* \backslash S$. *Then, the following holds:*

$$\frac{\|\boldsymbol{z}_M\|_2^2}{k^*} - \frac{\epsilon}{k - \widetilde{k}} \le \frac{\|\boldsymbol{z}_{S \backslash S^*}\|_2^2}{k - \widetilde{k}},$$

*where* $\widetilde{k} = O(k^* \log(\|\boldsymbol{w}^*\|_2/\epsilon))$.

*Proof.* Recall that the $k$ groups are added greedily to form $S = \text{supp}(\widehat{\boldsymbol{w}})$. Let $Q = \{i_1, i_2, \ldots, i_k\}$ be the $k$-groups selected when the greedy procedure (Algorithm 2) is applied to $\boldsymbol{z}$. Then,

$$\|\boldsymbol{z}_{G_{i_j} \setminus (\cup_{1 \leq \ell \leq j-1} G_{i_\ell})}\|_2^2 \geq \|\boldsymbol{z}_{G_i \setminus (\cup_{1 \leq \ell \leq j-1} G_{i_\ell})}\|_2^2, \quad \forall 1 \leq j \leq k, \; \forall i \notin Q.$$

Now, as $\cup_{1 \leq \ell \leq j-1} G_{i_\ell} \subseteq S, \; \forall 1 \leq j \leq k$, we have:

$$\|\boldsymbol{z}_{G_{i_j} \setminus (\cup_{1 \leq \ell \leq j-1} G_{i_\ell})}\|_2^2 \geq \|\boldsymbol{z}_{G_i \setminus S}\|_2^2, \quad \forall 1 \leq j \leq k, \; \forall i \notin Q.$$

Let $\text{G-supp}(\boldsymbol{w}^*) = \{\ell_1, \ldots, \ell_{k^*}\}$. Then, adding the above inequalities for each $\ell_j$ s.t. $\ell_j \notin Q$, we get:

$$\|\boldsymbol{z}_{G_{i_j} \setminus (\cup_{1 \leq \ell \leq j-1} G_{i_\ell})}\|_2^2 \geq \frac{\|\boldsymbol{z}_{S^* \setminus S}\|_2^2}{k^*}, \tag{15}$$

where the above inequality also uses the fact that $\sum_{\ell_j \in \text{G-supp}(\boldsymbol{w}^*), \ell_j \notin Q} \|\boldsymbol{z}_{G_{\ell_j} \setminus S}\|_2^2 \geq \|\boldsymbol{z}_{S^* \setminus S}\|_2^2$.

Adding (15) $\forall \, (\widetilde{k} + 1) \leq j \leq k$, we get:

$$\|\boldsymbol{z}_S\|_2^2 - \|\boldsymbol{z}_B\|_2^2 \geq \frac{k - \widetilde{k}}{k^*} \cdot \|\boldsymbol{z}_{S^* \setminus S}\|_2^2, \tag{16}$$

where $B = \cup_{1 \leq j \leq \widetilde{k}} G_{i_j}$.

Moreover using Lemma 2.2 and the fact that $|\text{G-supp}(\boldsymbol{z}_{S^*})| \leq k^*$, we get: $\|\boldsymbol{z}_B\|_2^2 \geq \|\boldsymbol{z}_{S^*}\|_2^2 - \epsilon$. Hence,

$$\frac{\|\boldsymbol{z}_M\|_2^2}{k^*} \leq \frac{\|\boldsymbol{z}_S\|_2^2 - \|\boldsymbol{z}_B\|_2^2}{k - \widetilde{k}} \leq \frac{\|\boldsymbol{z}_S\|_2^2 - \|\boldsymbol{z}_{S^*}\|_2^2 + \epsilon}{k - \widetilde{k}} \leq \frac{\|\boldsymbol{z}_{S \setminus S^*}\|_2^2 + \epsilon}{k - \widetilde{k}}. \tag{17}$$

Lemma now follows by a simple manipulation of the above given inequality. $\qquad\square$

## C  Proof of Lemma 3.3

*Proof.* Note that,

$$\|X\boldsymbol{w}\|_2^2 = \sum_i (\boldsymbol{x}_i^T \boldsymbol{w})^2 = \sum_i (\boldsymbol{z}_i^T \Sigma^{1/2} \boldsymbol{w})^2 = \|Z\Sigma^{1/2}\boldsymbol{w}\|_2^2,$$

where $Z \in \mathbb{R}^{n \times p}$ s.t. each row $\boldsymbol{z}_i \sim N(0, I)$ is a standard multivariate Gaussian. Now, using Theorem 1 of [4], and using the fact that $\Sigma^{1/2}\boldsymbol{w}$ lies in a union of $\binom{M}{k}$ subspaces each of at most $s$ dimensions, we have $\left(w.p. \geq 1 - 1/(M^k \cdot 2^s)\right)$:

$$\left(1 - \frac{4}{\sqrt{C}}\right) \|\Sigma^{1/2}\boldsymbol{w}\|_2^2 \leq \frac{1}{n}\|Z\Sigma^{1/2}\boldsymbol{w}\|_2^2 \leq \left(1 + \frac{4}{\sqrt{C}}\right) \|\Sigma^{1/2}\boldsymbol{w}\|_2^2.$$

The result follows by using the definition of $\sigma_{\min}$ and $\sigma_{\max}$. $\qquad\square$

## D  Proof of Theorem 3.4

*Proof.* Recall that $\boldsymbol{g}_t = \boldsymbol{w}_t - \eta \nabla f(\boldsymbol{w}_t)$, $\boldsymbol{w}_{t+1} = P_k^{\mathcal{G}}(\boldsymbol{g}_t)$. Similar to the proof of Theorem 3.1 (Appendix B), we define $S_{t+1} = \text{supp}(\boldsymbol{w}_{t+1})$, $S_t = \text{supp}(\boldsymbol{w}_t)$, $S_* = \text{supp}(\boldsymbol{w}^*)$, $I = S_{t+1} \cup S_*$, $J = I \cup S_t$, and $M = S_* \setminus S_{t+1}$. Also, note that $|\text{G-supp}(I)| \leq k + k^*$, $|\text{G-supp}(J)| \leq 2k + k^*$.

Now, using Lemma D.1 with $\boldsymbol{z} = (\boldsymbol{g}_t)_I$, we have: $\|(\boldsymbol{w}_{t+1} - \boldsymbol{g}_t)_I\|_2^2 \leq \frac{k^*}{k} \cdot \|(\boldsymbol{w}^* - \boldsymbol{g}_t)_I\|_2^2$. This follows from noting that $M = k + k*$ here. Now, the remaining proof follows proof of Theorem 3.1

closely. That is, using the above inequality with triangle inequality, we have:

$$\|\boldsymbol{w}_{t+1} - \boldsymbol{w}^*\|_2$$

$$\leq \left(1 + \sqrt{\frac{k^*}{k}}\right) \cdot \|(\boldsymbol{w}^* - \boldsymbol{g}_t)_I\|_2$$

$$\overset{\zeta_1}{\leq} \left(1 + \sqrt{\frac{k^*}{k}}\right) \cdot \|(\boldsymbol{w}^* - \boldsymbol{w}_t - \eta(\nabla f(\boldsymbol{w}^*) - \nabla f(\boldsymbol{w}_t)))_I\|_2 + 2\eta\|(\nabla f(\boldsymbol{w}^*))_{S_{t+1}}\|_2,$$

$$\overset{\zeta_2}{\leq} \left(1 + \sqrt{\frac{k^*}{k}}\right) \cdot \|(I - \eta H_{J,J}(\alpha))(\boldsymbol{w}_t - \boldsymbol{w}^*)_J\|_2 + 2\eta\|(\nabla f(\boldsymbol{w}^*))_{S_{t+1}}\|_2,$$

$$\overset{\zeta_3}{\leq} \left(1 + \sqrt{\frac{k^*}{k}}\right) \cdot \left(1 - \frac{\alpha_{2k+k^*}}{L_{2k+k^*}}\right)\|\boldsymbol{w}_t - \boldsymbol{w}^*\|_2 + \frac{2}{L_{2k+k^*}}\|(\nabla f(\boldsymbol{w}^*))_{S_{t+1}}\|_2, \tag{18}$$

where $\alpha = c\boldsymbol{w}_t + (1-c)\boldsymbol{w}^*$ for a $c > 0$ and $H(\alpha)$ is the Hessian of $f$ evaluated at $\alpha$. $\zeta_1$ follows from triangle inequality, $\zeta_2$ follows from the Mean-Value theorem and $\zeta_3$ follows from the RSC/RSS condition and by setting $\eta = 1/L_{2k+k^*}$.

The theorem now follows by setting $k = 2 \cdot \left(\frac{L_{2k+k^*}}{\alpha_{2k+k^*}}\right)^2$. $\qquad\square$

**Lemma D.1.** *Let $\boldsymbol{z} \in \mathbb{R}^p$ be such that it is spanned by $M$ groups and let $\widehat{\boldsymbol{w}} = P_k^{\mathcal{G}}(\boldsymbol{z})$, $\boldsymbol{w}^* = P_{k^*}^{\mathcal{G}}(\boldsymbol{z})$ where $k \geq k^*$ and $\mathcal{G} = \{G_1, \ldots, G_M\}$. Then, the following holds:*

$$\|\widehat{\boldsymbol{w}} - \boldsymbol{z}\|_2^2 \leq \left(\frac{M-k}{M-k^*}\right)\|\boldsymbol{w}^* - \boldsymbol{z}\|_2^2.$$

*Proof.* Let $S = \text{supp}(\widehat{\boldsymbol{w}})$ and $S_* = \text{supp}(\boldsymbol{w}^*)$. Since $\widehat{\boldsymbol{w}}$ is a projection of $\boldsymbol{z}$, $\widehat{\boldsymbol{w}}_S = \boldsymbol{z}_S$ and $0$ otherwise. Similarly, $\boldsymbol{w}^*_{S_*} = \boldsymbol{z}_{S_*}$. So, to prove the lemma we need to show that:

$$\|\boldsymbol{z}_{\overline{S}}\|_2^2 \leq \left(\frac{M-k}{M-k^*}\right)\|\boldsymbol{z}_{\overline{S_*}}\|_2^2. \tag{19}$$

We first construct a group-support set $A$: we first initialize $A = \{B\}$, where $B = \text{supp}(\boldsymbol{w}^*)$. Next, we iteratively add $k - k^*$ groups greedily to form $A$. That is, $A = A \cup A_i$ where $A_i = \text{supp}(P_1^{\mathcal{G}}(\boldsymbol{z}_{\overline{A}}))$.

Let $\widetilde{\boldsymbol{w}} \in \mathbb{R}^p$ be such that $\widetilde{\boldsymbol{w}}_A = \boldsymbol{z}_A$ and $\widetilde{\boldsymbol{w}}_{\overline{A}} = 0$, where $\overline{A}$ denotes the complement of $A$. Also, recall that $\|\boldsymbol{z}_S\|_0^{\mathcal{G}} = \|\boldsymbol{z}_{\text{supp}(\widetilde{\boldsymbol{w}})}\|_0^{\mathcal{G}} \leq |A| = k$. Then, using the optimality of $\widehat{\boldsymbol{w}}$, we have:

$$\|\boldsymbol{z}_{\overline{S}}\|_2^2 \leq \|\boldsymbol{z}_{\overline{A}}\|_2^2. \tag{20}$$

Now,

$$\frac{\|\boldsymbol{z}_{\overline{B}}\|_2^2}{M-k^*} - \frac{\|\boldsymbol{z}_{\overline{A}}\|_2^2}{M-k} = \frac{1}{M-k^*}\|\boldsymbol{z}_{\overline{B}\backslash\overline{A}}\|_2^2 - \frac{k-k^*}{(M-k^*)(M-k)}\|\boldsymbol{z}_{\overline{A}}\|_2^2. \tag{21}$$

By construction, $\overline{B}\backslash\overline{A} = \cup_{i=1}^{k-k^*} A_i$. Moreover, $\overline{A}$ is spanned by at most $M-k$ groups. Since, $A_i$'s are constructed greedily, we have: $\|\boldsymbol{z}_{A_i}\|_2^2 \geq \frac{\|\boldsymbol{z}_{\overline{A}}\|_2^2}{M-k}$. Adding the above equation for all $1 \leq i \leq k-k^*$, we get:

$$\|\boldsymbol{z}_{\overline{B}\backslash\overline{A}}\|_2^2 = \sum_{i=1}^{k-k^*} \|\boldsymbol{z}_{A_i}\|_2^2 \geq \frac{k-k^*}{M-k}\|\boldsymbol{z}_{\overline{A}}\|_2^2. \tag{22}$$

Using (20), (21), and (22), we get: $\frac{\|\boldsymbol{z}_{\overline{B}}\|_2^2}{M-k^*} - \frac{\|\boldsymbol{z}_{\overline{S}}\|_2^2}{M-k} \geq 0$. That is, (19) holds. Hence proved. $\qquad\square$

# E   Proof of Theorem 4.1

First, we provide a general result that extracts out the key property of the approximate projection operator that is required by our proof. We then show that Algorithm 3 satisfies that property.

In particular, we assume that there is a set of supports $\mathcal{S}_{k^*}$ such that $\mathrm{supp}(\boldsymbol{w}^*) \in \mathcal{S}_{k^*}$. Also, let $\mathcal{S}_k \subseteq \{0,1\}^p$ be s.t. $\mathcal{S}_{k^*} \subseteq \mathcal{S}_k$. Moreover, for any given $\boldsymbol{z} \in \mathbb{R}^p$, there exists an efficient procedure to find $S \in \mathcal{S}_k$ s.t. the following holds for all $S_* \in \mathcal{S}_{k^*}$:

$$\|\boldsymbol{z}_{S \setminus S_*}\|_2^2 \leq \frac{k^*}{k} \cdot \beta_\epsilon \|\boldsymbol{z}_{S_* \setminus S}\|_2^2 + \epsilon, \tag{23}$$

where $\epsilon > 0$ and $\beta_\epsilon$ is a function of $\epsilon$.

We now show that (23) holds for the SoG case, specifically Algorithm 3. For simplicity, we provide the result for non-overlapping case; for overlapping groups a similar result can be obtained by combining the following lemma, with Lemma B.2.

**Lemma E.1.** *Let* $\mathcal{G} = \{G_1, \ldots, G_M\}$ *be $M$ non-overlapping groups. Let* $\text{G-supp}(\boldsymbol{w}^*) = \{i_1^*, \ldots, i_{k^*}^*\}$. *Let $G$ be the groups selected using Algorithm 3 applied to $\boldsymbol{z} \in \mathbb{R}^p$ and let $S_i$ be the selected set of co-ordinates from group $G_i$ where $i \in G$. Let $S = \cup_i S_i$, and let $S_* = \cup_i (S_*)_i = \mathrm{supp}(\boldsymbol{w}^*)$. Also, let $G^*$ be the set of groups that contains $S_*$. Then, the following holds:*

$$\|\boldsymbol{z}_{S \setminus S^*}\|_2^2 \leq \max\left(\frac{k_1^*}{k_1}, \frac{k_2^*}{k_2}\right) \cdot \|\boldsymbol{z}_{S^* \setminus S}\|_2^2.$$

*Proof.* Consider group $G_i$ s.t. $i \in G \cap G^*$. Now, in a group we just select elements $S_i$ by the standard hard thresholding. Hence, using Lemma 1 from [10], we have:

$$\|\boldsymbol{z}_{(S_*)_i \setminus S}\|_2^2 \geq \frac{k_2}{k_2^*} \|\boldsymbol{z}_{S \setminus (S_*)_i}\|_2^2, \forall i \in G \cap G^*. \tag{24}$$

Due to greedy selection, for each $G_i, G_j$ s.t. $i \in G \setminus G^*$ and $j \in G^* \setminus G$, we have:

$$\sum_{i \in G \setminus G^*} \|\boldsymbol{z}_{S_i}\|_2^2 \geq \frac{|G \setminus G^*|}{|G^* \setminus G|} \sum_{j \in G^* \setminus G} \|\boldsymbol{z}_{S_j}\|_2^2.$$

That is,

$$\sum_{i \in G \setminus G^*} \|\boldsymbol{z}_{S_i}\|_2^2 \geq \frac{k_1}{k_1^*} \sum_{j \in G^* \setminus G} \|\boldsymbol{z}_{S_j}\|_2^2. \tag{25}$$

The lemma now follows by adding (24) and (25), and rearranging the terms. $\qquad\square$

Now, we prove Theorem 4.1

*Proof.* Theorem follows directly from proof of Theorem 3.1, but with (12) replaced by the following equation:

$$\|(\boldsymbol{w}_{t+1} - \boldsymbol{g}_t)_I\|_2^2 = \|(\boldsymbol{g}_t)_M\|_2^2 \overset{\zeta_1}{\leq} \frac{k^*}{k} \cdot \beta_\epsilon \|(\boldsymbol{g}_t)_{S_{t+1} \setminus S_*}\|_2^2 + \epsilon \overset{\zeta_2}{\leq} \frac{k^*}{k} \cdot \beta_\epsilon \cdot \|(\boldsymbol{w}^* - \boldsymbol{g}_t)_I\|_2^2 + \epsilon, \tag{26}$$

where $\zeta_1$ follows from the assumption given in the theorem statement. $\zeta_2$ follows from $\boldsymbol{w}^*_{S_{t+1} \setminus S_*} = 0$. $\qquad\square$

# F   Results for the Least Squares Sparse Overlapping Group Lasso

Lemma E.1 along with Theorem 4.1 shows that for SoG case, we need to project onto more than (than $k_1^*$) groups *and* more than (than $k_2^*$) number of elements in each group. In particular, we select $k_i \approx (\frac{L_{2k+k^*}}{\alpha_{2k+k^*}})^2 k_i^*$ for both $i = 1, 2$.

Combining the above lemma with Theorem 4.1 and a similar lemma to Lemma 3.3 also provides us with sample complexity bound for estimating $\boldsymbol{w}^*$ from $(y, X)$ s.t. $y = X \boldsymbol{w}^* + \boldsymbol{\beta}$. Specifically, the sample complexity evaluates to $n \geq \kappa^2 \left( k_1^* \log(M) + \kappa^2 k_1^* k_2^* \log(\max_i |G_i|) \right)$.

| Signal | IHT | GOMP | CoGEnT |
|---|---|---|---|
| Blocks | **.00029** | .0011 | .00066 |
| HeaviSine | .0026 | .0029 | **.0021** |
| Piece-Polynomial | **.0016** | .0017 | .0022 |
| Piece-Regular | .0025 | .0039 | **.0015** |

Table 1: MSE on standard test signals using IHT with full corrections

# G    Additional Experimental Evaluations

**Noisy Compressed Sensing:**    Here, we apply our proposed methods in a compressed sensing framework to recover sparse wavelet coefficients of signals. We used the standard "test" signals (Table 1) of length $2048$, and obtained $512$ Gaussian measurements. We set $k = 100$ for IHT and GOMP. IHT is competitive (in terms of accuracy) with the state of the art in convex methods, while being significantly faster. Figure 3 shows the recovered blocks signal using IHT. All parameters were picked clairvoyantly via a grid search.

Figure 3: Wavelet Transform recovery of 1-D test signals. (Left) The 'blocks' signal and recovery using IHT + Greedy projections. (Right) Objective function vs iterations on the 'blocks' signal.