[Reviews · NeurIPS 2016]

Reviewer 1

Summary

This paper is focused on solving sparse regression problem under exact group sparsity constraint using IHT-style algorithms. For every iteration of IHT, in order to tackle the generally NP-hard projection for overlapping group sparsity, a submodular optimization procedure is proposed. The recovery guarantees of the IHT algorithm with submodular optimization steps are established. Overall the paper is clearly written and the technical quality is good, but the novelty might be limited since the theoretical result has been established for IHT under standard unstructured sparsity assumption in [11]

Qualitative Assessment

In Section 1, the author gives the problem setting and review the existing approaches and results, with a clear identification of the current theoretical limitations and difficulties. In Section 2, it is nice to see that algorithms are presented in the boxes, which is easier for reader to follow. From the proof of Lemma 2.2, the approximation quality of submodular optimization to the NP-hard projection seems to be some well-studied property in submodular optimization community. The recovery guarantees presented in Section 3 and 4 are correct, but proof techniques are similar to those developed in [11].

Confidence in this Review

2-Confident (read it all; understood it all reasonably well)


Reviewer 2

Summary

The authors consider an \ell_0 penalized overlapping group regularization problem, and a corresponding greedy iterative hard thresholding method. The main challenge is in the projection step encountered in the latter method. For this, the authors proposed to use submodular optimization to produce an approximate projection. Statistical analysis is provided for the resulting estimator.

Qualitative Assessment

I generally like the proposal and find the use of submodular optimization for the approximate projection to be interesting. But at the same time, I'm really not convinced that the proposal is actually useful, either theoretically, or practically. Various claims made throughout the paper are quite confusing to me, and this certainly doesn't help. You criticize other works at the bottom of page 2 for assuming RIP, and also earlier in lines 26-28, but then you go and assume RSC above Definition 1. Aren't these essentially very similar assumptions ...? And furthermore, all in all, it seems to me that your results are pretty comparable to those from convex methods, if not a little weaker. So I'm left wondering: what is the point, theoretically? From a practical perspective, I could see Algorithms 1 and 2 be appealing for their simplicity. I could also imagine them being efficient. But the experiments given in the paper are pretty limited. The same can be said of the one experiment in the supplement. I think a much more thorough analysis and comparison should be conducted (especially if the theory is about on par with existing theory). What are FW, CoGEnT, and GOMP? It's not helpful to have the reader guess, but rather it would be worth spending the time explaining what these are down, even just briefly. I can guess was GOMP is, but FW = Frank-Wolfe is just a generic algorithm that can be applied to a wide range of optimization problems so that is not helpful, and I have no idea what CoGEnT is. Are any of these (perhaps FW) being applied to the same problem you are considering in (4) (or (5))? Are any of these considering the convex analog of (4) (or (5))? I don't think so, and convex analogs should really be compared, especially given your high-level discussion throughout the paper comparing your results/approach to convex ones. Other specific comments: - Lines 23-25: "In practice, IHT methods tend to be significantly more scalable than the (group-)lasso style methods that solve a convex program. But, these methods require a certain projection operator which in general is NP-hard to compute..." I know what you mean to convey here, but it reads a little funny. You mean that IHT methods are scalable in practice, becuase they only approximately solve the problems that they are designed for. Exact solutions are NP-hard. - The notation in (1) is a little crazy. It is almost universal to use \beta as the regression parameter. You use it as the noise variable. And you use \lambda as the noise variance, which is also confusing, because this is often reserved as a regularization parameter (say, penalty parameter for the group lasso). Again it is almost universal to use \sigma. - I'm confused about the discussion at the top of page 2. First of all, in the 2nd paragraph, you say that the improvements in sample complexity from the group lasso over the lasso are only logarithmic, which means that \log{p} becomes \log{M}, with M being the number groups and p being the number of features. (An aside, I don't see this as a bad thing; the feature dependence is already logarithmic to begin with, so what more could you ask for?) But isn't this precisely analogous to what you are claiming in the 4th paragraph, with your IHT group sparsity results, over IHT standard sparsity results? Furthermore, I don't understand the point of paragraph 3 and how it relates to your own work that you describe in paragraph 4. You say in paragraph 3 that "Greedy, Iterative Hard Thresholding (IHT) methods have been considered for group sparse regression 46 problems, but ... the guarantees they provide are along the same lines as the ones that exist for convex methods." Then in paragraph 4, you say that "In this paper, we show that IHT schemes with approximate projections for the group sparsity problem yield much stronger guarantees." This doesn't really make sense as written; are you using different approximation schemes, is that the point? And also, the guarantees that you describe in paragraph 4 for your method still do sounds quite similar to those from convex programming...

Confidence in this Review

2-Confident (read it all; understood it all reasonably well)


Reviewer 3

Summary

This is a good Paper. The Paper can be accepted for NIPS 2016 with the following revisions: The references need to be modified with more recent ones.

Qualitative Assessment

This is a good Paper. The Paper can be accepted for NIPS 2016 with the following revisions: The references need to be modified with more recent ones.

Confidence in this Review

3-Expert (read the paper in detail, know the area, quite certain of my opinion)


Reviewer 4

Summary

The paper deals with the learning problem of solving high-dimensional regression problems for sparse grouping structure. The author proposes a natural and simple modification of the current iterative hard thresholding method by appealing to submodular optimization to substitute the exact and often NP hard projections and also approach with a greedy manner. The paper provides with some theoretical bounds of the method and shows the method provides fast convergence guarantees and is computationally efficient and scalable. The author points out that they can also do well on poorly conditioned data and tries to extend to sparse overlapping groups and hierarchy structure. It seems the new method requires a very large scale of dataset to achieve actual good accuracy.

Qualitative Assessment

In general, the paper is very well organized and it is really worth mentioning that the author summarizes and points out the key assumption and limitation of their theoretical result in the remark section, which could be helpful for readers only focusing on the paper instead of appendix. The overall idea of submodular optimization and greedy algorithm seems natural and reasonable to me, and the theoretical guarantees are neat and well developed. Still, for the method to be competitive and the error occurred from making this approximation to be neglectable, it requires a large dataset and still a not too ill conditioned dataset. It would be interesting to see the comparison between this method and forward group lasso selection kind of method, and maybe some extension beyond gaussian distribution assumption.

Confidence in this Review

2-Confident (read it all; understood it all reasonably well)


Reviewer 5

Summary

This paper deals with high dimensional linear regression with relevant predictors belonging to a small number of groups. They use Iterative Hard Thresholding procedure with an approximate greedy projection algorithm. In theory they showed prediction accuracy is guaranteed.

Qualitative Assessment

This paper establishes theoretical guarantee for group Iterative Hard Thresholding procedure. RSC/RSS condition reasonable and could be automatically satisfied for linear regression. Some questions about Full Corrections: do you have any theoretical guarantee after you incorporate full corrections in your IHT algorithm?

Confidence in this Review

2-Confident (read it all; understood it all reasonably well)


Reviewer 6

Summary

This paper proposed an algorithm to estimate the group sparse linear regression model with the greedy hard-thresholding framework. Some existing iterative hard thresholding (IHT) algorithm for the group sparse linear regression model requires certain exact projection operation, which is NP-hard. To overcome such challenges, the authors proposed to use approximate projection which uses greedy strategies. The authors showed that such greedy approximation strategy can provide accurate estimation both in theory and in practice. Additional correction step can also be applied to further improve the estimation accuracy. By presenting results on both synthetic data and real data, the authors demonstrated that the proposed algorithm gave an accurate estimation and have better runtime performance when compared with existing work.

Qualitative Assessment

The authors proposed an estimation algorithm for group sparse linear regression model via greedy hard-thresholding. Estimation bounds are proved theoretically, and the experiment results showed that the proposed algorithm converges faster than the compared existing approaches, and provide more accurate estimation of the model parameters, which seems promising in practice. However, there are also a lot of issues that prevent this paper to be accepted by NIPS: 1. The notations used in this paper is a little bit messy. A lot of symbols and acronyms are used without definition or explanation. For example, \kappa on page 2 line 34, G_j \subseteq [p], RIP-style on page 2 line 83, RSC/RSS on page 2 line 86, a_{G_i} in Equation (2), \preceq symbol in Definition 2.1, etc. Although I finally figured out the meaning of some of these notations by looking through cited references, using notations without definitions and explanations would make the paper difficult to understand, and make the reader confused. 2. Regarding the novelty of the paper, the authors applied the IHT framework, which seems to be a well-studied framework for sparse structure estimation. The greedy selection strategy also has been adopted in existing work of sparse and group sparse estimation as mentioned by the authors in sec. 1.1, and the application scenario of linear regression is also extensively studied. How do the authors justify their contributions? 3. Regarding the experiments, the results look promising. The proposed algorithm converges faster and thus requires less time to estimate the linear group sparse model on synthetic data. However, on the real data, the proposed algorithm does not perform well in terms of prediction errors (Fig. 2, right, only better than FW method). Although IHT with correction performs the best among the methods under consideration, it makes me think that the proposed greedy IHT algorithm does not estimate the model parameter well, but it is the correction step saves everything in the real data application scenarios. Also, the authors even did not make any explanations about the results of sparse overlapping group lasso (Fig. 2 middle). Then why did the author present such results in the paper? 4. some minor things: a) the labels and legends in Fig. 1 and 2 are too small and hard to see. b) The phase transition plots in Fig. 1 do not have a legend. Without a legend, it is difficult for the reviewers and future readers to understand the meaning of the colors used in the plots, and thus can not understand the results presented by the figures.

Confidence in this Review

2-Confident (read it all; understood it all reasonably well)